# Peptide Inhibitors of Kv1.5: An Option for the Treatment of Atrial Fibrillation

**DOI:** 10.3390/ph14121303

**Published:** 2021-12-14

**Authors:** Jesús Borrego, Adam Feher, Norbert Jost, Gyorgy Panyi, Zoltan Varga, Ferenc Papp

**Affiliations:** 1Department of Biophysics and Cell Biology, Faculty of Medicine, University of Debrecen, Egyetem ter 1, H-4032 Debrecen, Hungary; jesus.borrego@med.unideb.hu (J.B.); feher.adam@med.unideb.hu (A.F.); panyi@med.unideb.hu (G.P.); veze@med.unideb.hu (Z.V.); 2Department of Pharmacology and Pharmacotherapy, Faculty of Medicine, University of Szeged, 6725 Szeged, Hungary; jost.norbert@med.u-szeged.hu; 3Department of Pharmacology and Pharmacotherapy, Interdisciplinary Excellence Centre, University of Szeged, 6725 Szeged, Hungary; 4ELKH-SZTE Research Group for Cardiovascular Pharmacology, Eötvös Loránd Research Network, 6725 Szeged, Hungary

**Keywords:** Kv1.5, I_Kur_, peptide inhibitor, atrial fibrillation

## Abstract

The human voltage gated potassium channel Kv1.5 that conducts the I_Kur_ current is a key determinant of the atrial action potential. Its mutations have been linked to hereditary forms of atrial fibrillation (AF), and the channel is an attractive target for the management of AF. The development of I_Kur_ blockers to treat AF resulted in small molecule Kv1.5 inhibitors. The selectivity of the blocker for the target channel plays an important role in the potential therapeutic application of the drug candidate: the higher the selectivity, the lower the risk of side effects. In this respect, small molecule inhibitors of Kv1.5 are compromised due to their limited selectivity. A wide range of peptide toxins from venomous animals are targeting ion channels, including mammalian channels. These peptides usually have a much larger interacting surface with the ion channel compared to small molecule inhibitors and thus, generally confer higher selectivity to the peptide blockers. We found two peptides in the literature, which inhibited I_Kur_: Ts6 and Osu1. Their affinity and selectivity for Kv1.5 can be improved by rational drug design in which their amino acid sequences could be modified in a targeted way guided by in silico docking experiments.

## 1. Introduction

Ion channels are transmembrane proteins, which form a pore in the cell membrane for ions to pass through. Several classifications of ion channels are known, for example, based on selectivity, gating or amino acid sequence. The gating mechanism can be of many types, such as voltage, stretch, ligand or temperature. Voltage-gated ion channels (VGICs) form one of the largest groups. These ion channels are involved in a great variety of cellular functions, such as generation of action potentials (AP) in excitable cells or activation in numerous non-excitable cell types, such as lymphocytes and tumor cells. VGICs typically consist of four subunits (potassium channels) or four domains (calcium and sodium channels), each of which is made up of six transmembrane helices (S1–S6). The first four helices together (S1–S4) are called the voltage-sensing domain (VSD), while the rest (S5–S6) build up the pore (Figure 1). The voltage sensing response mostly comes from the movement of S4, which has net positive electric charge, originating from positively charged amino acids: arginines and lysines. In response to a membrane potential change, these proteins open their ion-selective pore through which ions move passively across the membrane, driven by the electro-chemical potential difference [1,2,3,4]. The discovery of voltage-sensing phosphatase (VSP) and the voltage-activated proton channel (Hv1) revealed that the VSD can exist independently from the ion-conducting pore [5,6,7].

Among VGICs, the voltage-gated potassium channels (Kv) form a large family with some 40 members [8]. They are highly selective for potassium ions over other cations and expressed in almost all cell types, including muscle cells, neurons and immune cells, playing active roles in a variety of cellular functions. Kv channels provide the outward cation currents required to terminate the AP in excitable cells and allow the membrane potential to return to a negative resting potential following an AP. Several Kv channels contribute to shaping of the AP in the heart. The currents produced by these channels in cardiomyocytes include: the “transient outward” potassium current (I_to1_); the delayed rectifier potassium channel currents, which are named based on the speed at which they activate: slowly activating (I_Ks_), rapidly activating (I_Kr_) and ultra-rapidly activating (I_Kur_) [9]. I_Kur_ is generated by the potassium current through the Kv1.5 channel. I_Kur_ is present in human atrial myocytes but not in the human ventricle [10]. Many studies have concluded that inhibition of I_Kur_ could prolong the AP duration (APD) of atrial fibrillation patients [11,12], and by this, it can terminate the fibrillation, indicating that Kv1.5 is a potential target for atrial fibrillation therapy [13,14,15,16,17].

## 2. Diseases Related to Kv1.5

The most well-known channelopathy associated with the Kv1.5 channel is atrial fibrillation (AF). Today, a plethora of mutations have already been identified as causes of AF. Among them there are loss of function (LOF) mutations (E375X, Y155C, D469E and P488S), which make the atrial action potential (AP) prolonged, and gain of function (GOF) mutations (E48G, A305T and D332H), which shorten the AP. In the former case, the prolongation of the AP and the effective refractory period (ERP) increases the probability of early afterdepolarizations (EADs). However, during GOF mutations, the shortening of the ERP will increase the excitability of the atrial tissue as a potential mechanism behind AF [18,19].

Besides atrial fibrillation, mutations of the Kv1.5 channel gene can result in various diseases. Remillard and colleagues identified 17 single-nucleotide polymorphisms of the Kv1.5 gene in pulmonary arterial hypertension (PAH) patients [20], which may contribute to the downregulation of KCNA5, causing the increase of the vascular tone. Fu and colleagues [21] found that in intrauterine growth retardation, while Kv1.5 expression was decreased, the tyrosine-phosphorylation of these channels was significantly increased. This process led to the proliferation of the pulmonary artery smooth muscle cells, which eventually resulted in the thickening of the pulmonary arterial wall, i.e., PAH. MacFarlane and Sontheimer [22] showed in astrocytes that Kv1.5 is associated with Src family protein tyrosine kinases, which are responsible for astrocyte proliferation. This connection between Kv1.5 and astrocyte proliferation stimulated numerous tumor-related studies. Preussat and colleagues [23] found high Kv1.5 expression in human gliomas, which was the most prominent in astrocytomas, moderate in oligodendrogliomas and low in glioblastomas. Bielanska and colleagues pointed out that in stomach, pancreatic and breast cancer, the high expression of Kv1.5 was due to the presence of infiltrating inflammatory cells [24]. However, in bladder, skin, ovary and lymph node cancers, Kv1.5 was highly expressed in the tumorigenic cells. According to Vallejo-Gracia and colleagues [25], Kv1.5 expression shows an inverse correlation with lymphoma aggressiveness; therefore, the level of this protein can be useful in prognosis, treatment and outcome prediction as well.

## 3. Atrial Fibrillation and Possible Pharmacological Treatments

AF is characterized by an irregular and often rapid heart rate. In people with AF, blood flow is significantly slowed in the atria, which may cause blood to pool, greatly increasing the chances of blood clot formation. When a piece of a clot breaks off, it can travel to the brain and cause a stroke, which is one of the most serious consequences of AF. However, blood clots may circulate to other organs as well, blocking blood flow and causing ischemia. AF is the most common serious abnormal heart rhythm and, as of 2020, affects more than 33 million people worldwide [26]. As of 2014, it affected about 2 to 3% of the population of Europe and North America [27]. Due to AF, some patients need to constantly take blood thinners (platelet aggregation inhibitors and/or anticoagulants) to prevent blood clots, which can be very costly and can have very serious side effects, such as bleeding or hemorrhagic stroke.

The possible pharmacological strategies for AF treatment are the following:-Development and improvement of existing antiarrhythmic agents: Amiodarone derivates, Multi-channel blockers, etc.-Atrial selective therapeutic agents (ARDA): I_Kur_ blocker; I_K,Ach_ blocker; I_Na_, I_Kr_ blockers-Upstream therapy agents, drugs affecting structural remodeling; inflammation; hypertrophy; oxidative stress; etc.,-Gap junction modulators: Antiarrhythmic peptides affecting connexins Cx40 and Cx43

The most promising strategy to treat AF that avoids ventricular proarrhythmic side effects is the development of drugs known as “atrial selective drugs”. This concept would exploit distinct differences in expression patterns of individual ion channels and their different contribution to refractoriness between atrial and ventricular myocytes. Such atrial specific targets would be the following three known atrial specific ionic currents: (a) the ultra-rapid delayed rectified potassium current (I_Kur_); (b) the acetylcholine-sensitive inward rectifier potassium current (I_K,ACh_); (c) the constitutively active I_K,ACh_ currents (i.e., which are active even in the absence of agonists at muscarinic receptors).

Inhibition of the ion flow through Kv1.5, i.e., blocking I_Kur_, eliminates a component of the repolarizing current during atrial AP, thus prolonging the duration of the AP [12]. Almost all of the known Kv1.5 blockers are exclusively small molecules [8,28,29,30]. Pharmaceutical companies have made great efforts to develop selective I_Kur_ blockers as new pharmacological agents against AF. As a result, many new I_Kur_ blockers have been developed and tested since the beginning of this century: AVE0118, XEN-D101, DPO-1, vernakalant, etc.

AVE0118 (Figure 2) is a biphenyl derivative developed by Sanofi-Aventis. AVE0118 blocks I_Kur_ at micromolar concentrations in both native human atrial cells and Kv1.5 channel systems. In addition to the blocking of I_Kur_, the drug also blocked I_to_ and I_K,ACh_ currents at a similar concentration range [31,32].

AVE0118 shortened APD and ERP in atrial tissue from patients in sinus rhythm (SR), whereas APD/ERP was only slightly prolonged in tissues from patients in AF [32]. This observation is consistent with a previous study with the non-selective I_Kur_ blocker 4-aminopyridine [33]. AVE0118 has not been published in clinical trials and it appears that its development as a potential antirrhythmic drug is likely to have been halted. However, the compound was recently proposed as a new pharmacological tool for the treatment of obstructive sleep apnea [34].

XEN-D0101 (Figure 3) is an experimental compound developed by a small R&D company (Xention Ltd., Cambridge, UK). A first clinical trial with XEN-D0103 did not reduce the burden of AF in patients with paroxysmal AF [35].

DPO-1 (Diphenylphosphine oxide, Figure 4). DPO-1 blocks I_Kur_ rate-dependently at nanomolar concentrations in isolated human atrial myocytes. DPO-1 blocks other currents (such as I_to_) at higher micromolar concentrations. Furthermore, DPO-1 induced prolongation of APD in AF plateau, elevation and shortening in SR only in human atrial tissue and not in the ventricle [36].

Vernakalant (RSD1235, Cardiome and Astellas, Figure 5) is the molecule in the most advanced phase of study. It was approved by the European authorities, but the FDA did not allow intravenous conversion of AF. Vernakalant inhibited I_Kur_ in a positive frequency-dependent manner [37,38]. However, in human atrial cardiomyocytes, its effects on I_to1_ are small. In human atrial preparations vernakalant suppresses upstroke velocity, suggesting relevant inhibition of I_Na_ [37,38], so it can be considered a multichannel inhibitor rather than a selective I_Kur_ blocker. Vernakalant has rapid offset kinetics at sodium channels, so it was not expected to cause proarrhythmia and conduction disturbances at low heart rates [39,40]. However, vernakalant slowed conduction velocity at physiological heart rate both in the atria and in the ventricles of human hearts, calling into question the atrial selectivity of the drug effect [41]. Numerous clinical studies have shown the safety and efficacy of vernakalant in the transformation of AF. The AVRO study (phase III clinical study) demonstrated that vernakalant has superior efficacy compared with amiodarone in the acute conversion of recent cardiac arrhythmias AF [42,43]. In another study, vernakalant was shown to be safe and effective in combination with electrical cardioversion [44] and was approved for clinical practice in the European Union in 2010 (but not in the US [45]).

## 4. Peptide Modulators of the Kv Channels

Animal venoms are a complex cocktail of oligopeptides, free amino acids, nucleotides, low molecular weight salts, organic compounds, peptides and proteins [46]. These venoms are employed for prey hunting and protection against predators [47]. In this complex mixture of bioactive molecules, the lethal toxin often represents only a minor proportion, along which many other non-lethal components with interesting bioactivities are present, which can be used for the development of pharmaceutical agents, insecticides and research tools in the characterization of ion channels [48].

Research on peptide modulators of Kv channels started in the 1980s [30]. To date, ~460 toxins have been reported exclusively for voltage-gated K^+^ channels (Figure 6), scorpion venoms being the major source of these molecules with 203 entries, followed by the spider venoms with 102 [49,50,51]. These arachnid venom-derived peptides interact with Kv channels in two different modes: either as pore blockers or as gating modifiers, with a very specific interaction with different regions of the ion channels.

### 4.1. Pore Blocker Peptides

All known scorpion toxins affecting potassium channels (KTx) physically occlude the channel pore, which makes them pore blockers [52]. Based on homology, cysteine pairing pattern and activity, KTxs have been classified into six families: α-KTx, β-KTx, γ-KTx, κ-KTx, δ-KTx [53] and ε-KTx (Figure 7).

The α-KTx family is the largest family, with 174 members grouped in 31 subfamilies, followed by the β-KTx family with 35 members grouped in 3 subfamilies and the γ-KTx family with 30 members grouped in 5 subfamilies [51,54,55,56]. All these three families share a common structural motif comprising one or two α-helices connected to a triple-stranded antiparallel β-sheet stabilized by three or four disulfide bonds (CSα/β) [55]. The majority of the members of the α-KTx family have been isolated from the venoms of scorpions of the *Buthidae* family [57]. These peptides range from 23 to 43 residues in size and recognize *Shaker*-type Kv channels and Ca^2+^-activated K^+^ channels [58]. β-KTx peptides are longer than the α-KTx, ranging from 45 to 75 residues in size. The difference between the sizes of these families can be explained by an N-terminus α-helix with cytolytic and/or antimicrobial activity, followed by the C-terminal region with a CSα/β motif that confers the K^+^ channels blocking activity [59]. Peptides of the γ-KTx family were discovered in the venom of scorpions of the genus *Centruroides*, *Mesobuthus* and *Buthus* [60]. Their length ranges from 36 to 47 residues, and they are described as mainly targeting K^+^ channels of the ERG (ether-á-go-go gene) family [61].

The κ-KTx family is comprised of 18 members grouped in 5 subfamilies. All these peptides have been isolated from scorpion venoms of the genus *Heterometrus* and *Opisthacanthus*. Peptides of this family consist of 22 to 28 amino acid residues and are considered weak inhibitors of K^+^ channels (all of them showing effect in the µM range). The structure of κ-KTx peptides is characterized by two parallel α-helices linked by two disulfide bridges (CSα/α) [62].

The δ-KTx family is comprised of 7 members grouped in 3 subfamilies, ranging from 59 to 70 residues, which have been isolated from the venom of scorpions of the genus *Hadrurus, Mesobuthus* and *Lychas*. The members of the δ-KTx family are characterized by a Kunitz-type fold, which is represented by two antiparallel β-strands and two, or more often, one helical region [63]. Moreover, δ-KTx members possess a dual activity inhibiting proteolytic enzymes (e.g., trypsin) in nanomolar concentration and blocking Kv channels [64].

The ε-KTx family is the smallest one. It is comprised of only two members (29 amino acid residues length), both of them isolated from the venom of the scorpion *Tityus serrulatus*. The structure of these peptides consists of an inhibitor cystine knot type scaffold (ICK). However, the structure is completely devoid of the classical secondary structure elements (α-helix and/or β-strand) [65].

In previous works, a seventh family of KTx is mentioned [57,66]. This family, known as λ-KTx, also presents an ICK scaffold, but unlike the ε-KTx family, the structure is a CSα/β fold. The best-characterized peptide of this family, the λ-MeuTx-1, showed a blocking effect in the *Shaker* K^+^ channel but not in other Kv channels [67]. However, nowadays, this peptide has been reclassified into the scorpion calcin-like family, which is why the λ-KTx family does not appear in the classification of the KTx anymore.

#### Mechanism of Action of the Pore Blockers

KTxs can interact with Kv channels through different mechanisms. However, three major mechanisms have been described. The first one is the so-called “functional dyad” model, which is the most frequently identified and the best characterized. The dyad is composed of two highly conserved amino acid residues. In the first position, there is a lysine and in the second position, a neighboring aromatic or aliphatic residue [18]. In this model, the β-sheet side of the toxin faces the entrance of the channel pore and the lysine side chain in the selectivity filter (Figure 8). The hydrophobic interaction of the second amino acid is involved mostly in the high-affinity binding [58]. The lysine side chain is attracted to the pore, where a ring of aspartate or glutamate residues surrounds it [68], while the aromatic or aliphatic residue might interact with a tyrosine or tryptophan residue of one of the channel α-subunits [69]. The second mechanism is called the “ring of basic residues”. This mechanism has been demonstrated for the entire α-KTx subfamily 5 and α-KTx4.2 interacting with the KCa2.x channels. In this model, a cluster of basic residues (2-4 Arg and Lys) interacts with residues of the channel situated at the turret and the bottom of the vestibule. However, this cluster is located in the α-helix of the toxin instead of the β-hairpin [69,70]. The last model involved the interaction of the γ-KTxs with the ERG channels. The binding occurs in a hydrophobic binding site comprising an amphipathic α-helix located in the S5-P linker and the P-S6 linker. In this interaction, the toxins bind in an off-center position in the outer vestibule; however, the lack of the Lys residue found in the functional dyad lead to a reduction but not a total occlusion of K^+^ current [71].

### 4.2. Gating Modifiers Peptides

In contrast to scorpion venom peptides, spider venom peptides are mostly gating modifiers. These peptides range from 29 to 35 residues and were isolated mainly from the venoms of the *Theraphosidae* family. They are characterized by a promiscuous selectivity between Ca^2+^, Na^+^ and K^+^ channels. For example, the Hanatoxin (HaTx1) [72] and HpTx1 [73] (patent) toxins show effect on Ca^2+^ and K^+^ channels. On the other hand, the VsTx1 [74], PaTx1 [75,76], HmTx1 [77,78] and GiTx1 [79] toxins have been reported as Na^+^ and K^+^ channel modulators. Furthermore, it has been discovered that in the toxin-channel interaction, the lipids in the cell membrane are also involved [80,81]. For Kv channels, the selectivity of these toxins becomes a little tighter since all of these peptides affect mainly Kv2 and/or Kv4 subfamilies [66]. Although, peptides as the GiTx1 [79] and JZTX-1 [82] have shown an effect on hERG channels.

Spider gating modifier peptides present an ICK scaffold, where the β-sheet typically comprises two β strands (a third strand in the N-terminal can be present sometimes) stabilized by a cysteine knot. This knot comprises a ring formed by two disulfides and the intervening polypeptide backbone, with a third disulfide bridge going through the ring to create a pseudo-knot (Figure 9) [83]. The ICK turns these peptides into hyperstable proteins with tremendous chemical, thermal and biological stability [84].

#### Mechanism of Action of the Gating Modifiers

Spider gating modifier peptides bind to a region of the channel that changes conformation during gating and influences the gating mechanism by altering the relative stability of closed, open or inactivated states [85]. Most of the peptides possess a cluster of solvent-exposed hydrophobic residues (hydrophobic patch) surrounded by highly polar residues (charge belt), enhancing the affinity for the Kv channels by allowing the toxins to partition into the membrane [86]. These peptides interact with the Kv channel in the VSD region, specifically with the paddle motif, a mobile helix-turn-helix motif composed of the C-terminal portion of S3, and the S4 helix (Figure 8). The specific interactions have been revealed for several toxins. For example, in the binding between the HaTx1 and the Kv2.1 channel, the interaction with the F274 and E277 in the C-terminal portion of S3 plays a crucial role [87]. These molecular determinants are shared for the JZTX-XI toxin and the Kv2.1 channel interaction [88] and possibly for the interaction between the JZTX toxins and the hERG channels since the binding sites in the paddle motif of Kv2.1 (I273, F274 and E277) are conserved in the paddle motif of the hERG channel (I512, F513 and E518) [82]. On the other hand, despite HpTx2 binding to the same paddle motif in the Kv4 channel, HpTx2 binding does not require a charged amino acid for the interaction since hydrophobic residues (L275 and V276) are the most important for the binding [89]. Moreover, in experiments using chimera constructs in which the linker region S3-S4 of the Kv2.1 channel (TLTx1-insensitive) was replaced by the corresponding Kv4.2 domain, it was observed that the Kv2.1 channel became sensitive to TLTx1 [90]. These data suggest that even if toxins share the same binding site (paddle motif), the molecular determinants for the interaction can change between different families of Kv channels.

## 5. Osu1 and Ts6: The Known Peptide Modulators of the Kv1.5

To the best of our knowledge, only two peptides are known in the literature that have modulated the Kv1.5 ion current: Osu1 [91] and Ts6 [92,93]. Osu1 is a peptide isolated from the venom of a tarantula, called *Oculicosa supermirabilis*. It is a 64 amino acid peptide with a mass of 7478 Da, and its spatial structure is formed by four disulfide bridges (Figure 10) [91]. Electrophysiological recordings showed that the total venom of *Oculicosa supermirabilis*, as well as the native and recombinant Osu1, slowed the activation kinetics of the Kv1.5 current at ~μM peptide concentration. The slowing of the activation kinetics of the current was consistent with a ~40 mV shift in the conductance vs. membrane potential (G-V) relationship of the Osu1 bound channels, as compared to the toxin-free clontrol. In other words, the membrane potential at which 50% of the Kv1.5 channels are open (V_1/2_) is about 40 mV more positive when Osu1 is present. This rightward shift in the G-V indicates that Osu1 is most likely not bound to the pore of Kv1.5 but rather to the VSD, hindering its movement at depolarization; thus, Kv1.5 opens only at more positive voltages. As a result, in a certain membrane potential range, especially at mild depolarizations close to the activation threshold of the channel, the binding of Osu1 to the voltage sensor appears as an inhibitory effect. Based on this, it is possible to reduce the I_Kur_ current through Kv1.5 using Osu1 and thereby influence the shape and duration of the AP in the atrium of the heart.

Ts6 (α-KTx 12.1), previously known as butantoxin, also known as TsTX-IV, is the other known Kv1.5 modulating peptide. This peptide was isolated from the venom of the scorpion *Tityus serrulatus*. It consists of 40 amino acids with a molecular mass of 4506 Da and with 8 cysteine residues (Figure 11) [92,93,94]. Ts6 is primarily known as a peptide that inhibits Kv1.2 and Kv1.3 ion channels at nM concentrations (the IC_50_ values were 6.19 ± 0.35 nM for Kv1.2 and 0.55 ± 0.20 nM for Kv1.3) [92], but in the selectivity experiments, Ts6 also inhibited the current flowing through Kv1.5 at μM concentrations. Besides its effects on ion channels, Ts6 has also shown a pro-inflammatory effect by increasing the levels of cytokines, such as interleukin-6 (IL-6), both in in vivo and in vitro models [94,95]. IL-6 increase can cause cardiac or systemic inflammation, which in turn can rapidly lead to atrial electrical remodeling [96]. Thus, any other potential Kv1.5 blocking peptides should be tested for such adverse effects. After this report on the blocking effect of Ts6, no more results have been published in the literature that further investigated the inhibition of Kv1.5 by Ts6 or that any attempt had been made to utilize this knowledge in any way, such as in the treatment of atrial fibrillation.

## 6. Selectivity of Kv1.5 Inhibitors

Selectivity plays an important role in the potential therapeutic application of an ion channel blocker since the higher the selectivity, the lower the risk of side effects. As mentioned earlier, several small molecules targeting the Kv1.5 channel have been tested as candidates for the treatment of AF. In all of them, affinity and selectivity vary over a broad spectrum. AVE0118 exhibits an IC_50_ of 6.9 µM for the Kv1.5. However, in the same concentration range (10 µM), AVE0118 is able to block I_to_ and I_K,ACh_ [32]. The same is true for vernakalant, which is a multichannel blocker in the µM range [38]. XEN-D0103 inhibits Kv1.5 with an IC_50_ of 25 nM and shows more than 500-fold selectivity over hERG, Kv4.3, Nav1.5, Cav1.2 and Kir2.1 [97]. Similarly, DPO-1 shows a Kd value of 30 nM and blocks other channels only in the µM range [36].

Peptides in general have a much larger interacting surface with ion channels compared to small molecule inhibitors. Therefore, due to multiple points of contact, peptides potentially exhibit higher affinity and selectivity for channels with a lower chance of inhibition or modification of other ion channels than known small molecule inhibitors. For example, the peptide toxin Vm24 (α-KTx 23.1) shows high affinity (Kd = 2.9 pM) for the Kv1.3 channel and exhibits more than 1500-fold selectivity over other ion channels, including other Kv channels, KCa channels and Nav1.5 [98]. On the other hand, the small molecule tetraethylammonium (TEA) blocks numerous types of Kv channels, including members of the Kv1, Kv2 and Kv3 families, as well as KCa channels with a Kd value between 0.4 mM and 8 mM [99]. Pap-1 is a small molecule with one of the highest affinities for Kv channels: it blocks Kv1.3 in nM concentration (EC_50_ = 2 nM). PAP-1 is 23-fold selective over Kv1.5, 33- to 125-fold selective over other Kv1-family channels and 500- to 7500-fold selective over other K^+^, Na^+^, Ca^2+^ and Cl^−^ channels [100]. However, this nM affinity is still 1000 times lower compared to the pM affinity of the peptide Vm24. Another example is the toxin Gr1b. This peptide inhibits the Nav1.7 channel with a Kd value of 40 nM and exhibits 10- to 30-fold selectivity over other Nav channels [74]. In this case, the small molecule tetrodotoxin (TTX) shows a slightly higher affinity for Nav channels (Kd value of 4 to 25 nM). However, TTX can block different subtypes of Nav channels with almost the same Kd, showing a selectivity between 1.5- to 6-fold [101]. GX-674 small molecule inhibits Nav1.7 at subnanomolar concentration (Kd = 0.1 nM), but it is not selective over Nav1.6 and Nav1.2 [102].

So far, only Osu1 and Ts6 have been reported as Kv1.5 peptide inhibitors. Native Osu1 (peptide purified directly from venom) showed inhibitory activity at 0.9 µM, while recombinant Osu1 (expressed in bacteria) showed the same activity at 3 µM [91]. Nothing is yet known about the selectivity of Osu1. Ts6 inhibited Kv1.5 at a concentration of 1 µM. However, Ts6 also inhibited Kv1.2, Kv1.3 and Shaker channels with higher affinity (in nM) and Kv1.6, Kv7.2, Kv7.4 and hERG with similar affinity as Kv1.5 (in μM) [92]. One of the main reasons for the lack of peptide inhibitors for Kv1.5 is the presence of a positively charged Arg residue (R487) in the pore region of the channel, a feature missing in other Kv channels [103,104], which prevents Kv channel pore blocker peptides from binding to Kv1.5. The mutation of this Arg to Val (R487V) or Tyr (R487Y) made Kv1.5 sensitive to BgK, a known inhibitor toxin for Kv1 channels from the sea anemone *Bunodosoma granulifera* [103,105,106]. Knowing the binding mechanism of other toxins to Kv channels and the particular properties of the Kv1.5 channel, the selectivity of Ts6 and Osu1 (and other peptides) and their affinity for Kv1.5 could be improved by rational drug design in which their amino acid sequences are modified in a targeted way guided by in silico docking experiments.

## 7. Improving Selectivity and Affinity of Peptide Toxins

To improve the selectivity and the affinity of peptide toxins, it is necessary to understand well the properties of peptides isolated from animal venoms; however, some barriers must be overcome. Most of the time, it is difficult to isolate or study a peptide from the extremely limited amount of venom that can be obtained from the animals. This problem can be solved by chemical synthesis or recombinant expression of these peptides. After appropriate standardization of the expression protocols, a sufficient amount of peptide can be obtained [105,106,107] to fully characterize the biological activity of the peptides. However, as mentioned earlier, most toxin peptides are rich in disulfide bonds, resulting in low yields of the bioactive product with the desired disulfide bridge configuration.

### 7.1. Achieving the Native Peptide Scaffold

The primary, secondary and tertiary structure of the peptides are crucial to their specificity and functionality. Generally, methods to elucidate the peptide structure, such as NMR and X-ray studies, require an amount of toxin that is difficult to obtain from the natural source. Recombinant expression and chemical synthesis of peptides are excellent tools to reach the required amounts of the toxin. There are different recombinant expression systems to produce the target peptide. To date, *Escherichia coli* (*E. coli*) represents the most widely used heterologous expression system in which recombinant peptides are usually accumulated in the cytoplasm. However, because of the disulfide bonds, they tend to be misfolded and aggregate [108]. To solve this problem, some alternatives have been developed: (1) several *E. coli* strains have been genetically modified to generate strains in which the reducing cytoplasmic environment is more favorable for disulfide bond formation [109,110], (2) use of vectors with a signal sequence to take the protein into the periplasmic space with a more oxidizing environment and proteins that catalyze and rearrange the disulfide bonds [111], and (3) co-expression of the peptide along with fusion proteins or chaperones that enhance proper folding [112]. If the *E. coli* system does not work, yeast is the next option. These cells are used to produce recombinant proteins that are not produced well in *E. coli* because of folding issues or the need for glycosylation. Yeasts are easier and cheaper to work with than insect or mammalian cells and are easily adapted to fermentation processes. The two most commonly used yeast strains are *S. cerevisiae* and *P. pastoris* [107,113]. Although *E. coli* and yeast are the most commonly used expression systems for animal toxin production, other systems such as insect cells have also been reported [114,115].

On the other hand, there is solid-phase peptide synthesis (SPPS), which is an efficient method for producing peptides and small proteins. Some important advantages are that this approach allows the incorporation of non-native elements, such as N-substituted and D-amino acids, and the replacement of the backbone amide bonds. SPPS can also be used to generate peptides that cannot be produced by expression systems because they are toxic [116]. A disadvantage of SPPS over recombinant expression of toxins is that SPPS requires extensive screening of in vitro folding conditions, which can be further complicated because many toxins have multiple disulfide bonds [117]. However, it is worth noting that if the native disulfide pattern of the peptide is known, Cys with protecting groups can be used. These Cys protecting groups can be selectively removed to create a bond between two specific Cys in the structure, resulting in a peptide with the same Cys framework as the native peptide [118].

### 7.2. Uncovering Amino Acids Involved in Selectivity and Affinity

Once the peptide (recombinant or synthetic) has been produced with structural and biological properties similar to the native one, selectivity and affinity enhancement can be performed. The first step would be to know the amino acids involved in the interaction of the peptide with the channel. Solving the structure of the peptide–channel complex would reveal the amino acids that bind directly to the channel. Banerjee et al. [68] have used X-ray crystallography to study the structure of the complex formed by charybdotoxin (ChTx) (α-KTx 1.1) and a chimeric version of a voltage-gated potassium channel formed by Kv2.1 and Kv1.2. They confirmed the occlusion of the channel pore by Lys27 and showed the interactions between the amino acids of the toxin and the amino acids located in the mouth of the channel pore. Similarl to X-ray crystallography, cryo-electro microscopy was used to investigate ion channel–peptide toxin complex with Nav1.7 and ProTx2 [119,120]. Although these approaches can provide accurate information, obtaining the crystal could be difficult. For this reason, basic alanine scanning has become a widely used strategy for identifying side chains that play a key role in toxin–channel interactions. By mutating each native amino acid in the primary sequence to alanine one at a time, the importance of each amino acid in the binding interaction can be revealed [121]. Then, the amino acids directly involved in the binding interaction can be further optimized to improve the overall potency and selectivity of the peptides [122]. Double-mutant cycle analysis is another method to measure the strength of intermolecular pairwise interactions in protein–ligand and protein–protein complexes [123]. However, these techniques are time-consuming processes that become less practical with increasing size of the peptides under investigation. Therefore, computational methods are valuable tools to construct accurate models of toxin–channel complexes. Docking methods and molecular dynamics simulations can be used to find valuable hints to identify amino acid residues that need to be mutated to achieve the desired selectivity and then calculate the free energy perturbation between the native toxin and its analogs to evaluate the effects on binding of each mutant [124]. In these docking approaches, multiple related peptides can be tested in one specific channel to determine how structure and small changes in amino acid sequence affect the binding interaction. Docking analysis of charybdotoxin (ChTx) and margatoxin (MgTx) in the native version and the mutant version of Kv1.3 revealed not only the difference between the interaction of the two toxins but also how the toxins were reoriented to block the mutant channel, which helped to explain the results observed in patch-clamp [125]. On the other hand, it is possible to use docking to analyze the selectivity of one toxin for multiple channels. In this way, it was explained why the toxin Css20 (α-KTx 2.13) is more selective for channels Kv1.2 and Kv1.3 than for Kv1.1 and Kv1.4 [126].

With these methodologies, the rational design of more potent and selective peptides can be done. For example, MeKTx13-3 (α-KTx 3.19) is a toxin with a promiscuous effect on Kv1.1, Kv1.2, Kv1.3 and Kv1.6 channels, but the mutation or addition of specific residues improved either selectivity against Kv1.3 [127] or affinity for the Kv1.1 channel [128]. Similarly, the [N17A/F32T]-AnTx (an analog of AnTx, α-KTx 6.12) showed a 16,000-fold increase in selectivity towards the Kv1.3 channel while maintaining the high affinity of the native peptide for the channel [129]. Another approach to find amino acids that can be used as targets for improving selectivity and affinity is to analyze the sequence of related toxins. For example, the toxin OdK1 (α-KTx 8.5) differs from OSK3 (α-KTx 8.8) only by two C-terminal residues but shows a pronounced preference for Kv1.2, implying that these two amino acid residues are involved in the selectivity for the channel [130]. However, it is not only amino acid changes that can affect toxin activity. It has been shown that amidation in the C-terminus of urotoxin (α-KTx 3.19) increases its potency towards the Kv1.2 channel [131]. Moreover, several scorpion toxins have been shown to require C-terminus amidation for full biological activity, without which potency is severely reduced [132].

The above techniques attempt to improve the affinity and selectivity of peptides by specific changes in the amino acid sequence. However, peptides with blocking activity resulting from random recombination of related protein regions have also been reported. This recombination creates libraries of more than 1,000,000 different peptides that are tested by phage display to select chimeras that show interaction with the desired channels [133]. Similarly, fragment-based drug discovery is an excellent technique for discovering drugs. This approach first identifies starting points as small molecules. Then, these fragments are expanded or linked together to generate drugs. Although fragments bind to proteins with relatively low affinity, they form high quality binding interactions with the protein as they overcome a significant entropy barrier to bind [134,135].

In addition, there are other methods and techniques for increasing the affinity and selectivity of a peptide. For example: acidic-residue-function-guided drug design; chemical modification; residue truncation; binding interface modulation; reducing conformational flexibility; scaffold-/target-biased strategies; Artificial Intelligence-guided drug design [136,137,138].The overall methodological background and information on the interaction between toxins and Kv channels allows us to work with toxins, such as Osu1 and Ts6, generating and testing analogs until the toxin with the best affinity and selectivity is found, or to find analogs of other toxins related to Osu1 or Ts6 that may have similar activity on the Kv1.5 channel.

## 8. Testing Kv1.5 Modulation in AF Models

Different AF models exist to investigate the characteristics and parameters of AF and to test potential drug candidates. One of these is the various cell lines that have been created specifically for this purpose. An atrial muscle cell-derived cell line (iAM, immortalized atrial myocyte) has been developed from rat cells [139]. In their work, the authors performed RT-qPCR analysis of iAMs in different stages of differentiation and showed a rapid increase in mRNA levels of cardiac transcription factors, ion channels, Ca^2+^-handling proteins and sarcomeric proteins. iAMs acquired properties of atrial rather than ventricular myocytes, and a monolayer cell culture created from the iAM was developed as an in vitro model of AF. In this model, a spiral wave can be generated that returns to and maintains itself as re-entrant circuit, using optical voltage mapping and high-frequency (10–50 Hz) electrical point stimulation. The resulting re-entrant circuits could be terminated by prolonging the APD of the iAMs using a known K^+^ channel inhibitor, tertiapin (a peptide isolated from honeybee venom, Kir3.x-specific inhibitor). Based on these results, this AF model system seems to be suitable to test other K^+^ channel inhibitors and modulators.

However, before testing a drug candidate molecule in this AF model, it is useful to test it on individual cells of cell lines focusing on the action potential parameters, such as AP duration (APD_90_), AP amplitude, the maximum speed of depolarization (V_max_) and plateau potential at the point corresponding to 50% duration of APD_90_ (plateau50). If these listed parameters are not affected by the molecule to be tested, it is not expected to affect the AF model either. The most important parameter is APD_90_: if prolonged, there is a good chance that fibrillation will stop in the AF model. The same group is currently working on creating a human version of the iAM cell line (hiAM) [140].

A similar cell line AF model was developed by Peter H. Backx’s group. They used human embryonic stem cells (hESCs) to generate atrial-like cardiomyocytes (CMs) and to create an AF model for pharmacological testing. Using optical mapping techniques, atrial-like confluent CM cells showed uniform AP propagation and rapid re-entrant rotor patterns. They tested anti-arrhythmic drugs (flecainide, dofetilide in μM concentration) on single cells and cell sheets. Flecainide profoundly slowed upstroke velocity without affecting AP duration, while dofetilide prolonged APs and reduced cycle lengths of rotors in cell sheets [141].

LOF mutations in the PITX2 (specifically expressed in the left human atrium) transcription factor gene have been shown to cause familial AF. Boris Greber’s group generated a PITX2-deficient cell line to model AF and unravel PITX2-regulated downstream genes for drug target discovery. Their F1 cells were capable of spontaneously differentiating into cardiomyocytes; moreover, all cell lines could selectively be differentiated in a cardiac subtype-specific manner, i.e., form atrial or ventricular cardiomyocytes.

On the other hand, the anatomical and physiological similarities between humans and animals, particularly mammals, allow researchers to study a variety of mechanisms and novel therapies in animal models [142]. Despite AF being quite common in humans, spontaneously occurring AF has only been reported in relatively few animal species, such as cats, dogs, pigs, goats, sheep, cattle, horses, camelids and monkeys. In horses and cattle, the prevalence of spontaneous AF is the highest (about 2.5%), whereas in the other animals spontaneous AF cases rarely occur as an isolated problem without other cardiac diseases [143]. For this reason, AF has to be induced in the animal model through rate-related electrical remodeling or with atrial-structural remodeling [144] using models such as rapid atrial tachypacing, heart failure-associated AF and vagal tone-induced AF [143]. In the following section, we will focus on experiments in which the effects of drugs targeting I_Kur_ have been studied in animal models. However, we refer readers to two reviews by Shuttler et al. (2020) [145] and Saljic et al. (2021) [143], where they discussed extensively the animals models for AF studies.

Efforts by pharmaceutical companies to find new selective I_Kur_ blockers as novel pharmacological agents against AF have generated new data on the effects of these compounds in animal models. Some of these drugs are AVE0118, XEND101, DPO-1 and vernakalant [146], mentioned above. AVE0118 was tested in dogs and goats, showing fully restored atrial contraction without proarrhythmic effects on the ventricle [147,148]. Moreover, AVE0118 in combination with dofetilide or ibutilide showed effective cardioversion in persistent AF [149]. AVE1231 showed similar results in pigs and goats, prolonging atrial refractoriness with no effects on ECG intervals and ventricular repolarization [150]. In another study in dogs, XEN-D0101 and XEN-D0103 selectively blocked I_Kur_ current and prolonged the atria effective refractory period and decreased the duration of AF [151,152]. The same results were observed when the compound DPO-1 was tested in dogs [153]. Vernakalant (RSD1235) has been approved in European countries for acute cardioversion of AF with recent onset. It has been tested in several animal models, including goats [154], pigs [155] and dogs [156]. However, nowadays, it is considered as a multichannel blocker rather than an I_Kur_ specific blocker [143].

Although none of the above cases involved experiments with peptides, it is clear that the use of cell lines and animal models for AF is a potential method for testing peptide inhibitors to discover and develop new drugs for the treatment and prevention of AF.

## 9. Concluding Remarks

The technology to make peptides more selective for a given ion channel is known, along with the computer modelling to aid design, and there are excellent tools to test the efficacy of the peptides. The selectivity of ion channel inhibitors is extremely important for future therapeutic application in order to reduce the unwanted side effects—the higher the selectivity, the lower the risk of side effects. In the literature, there are only two peptides, Osu1 and Ts6, which can bind to the Kv1.5 channel. They are the candidates to elucidate the mechanisms of interaction between peptides and the Kv1.5 channel. Improving their selectivity for I_Kur_ can serve as an option for the treatment of atrial fibrillation.

## Figures and Tables

**Figure 1 pharmaceuticals-14-01303-f001:**
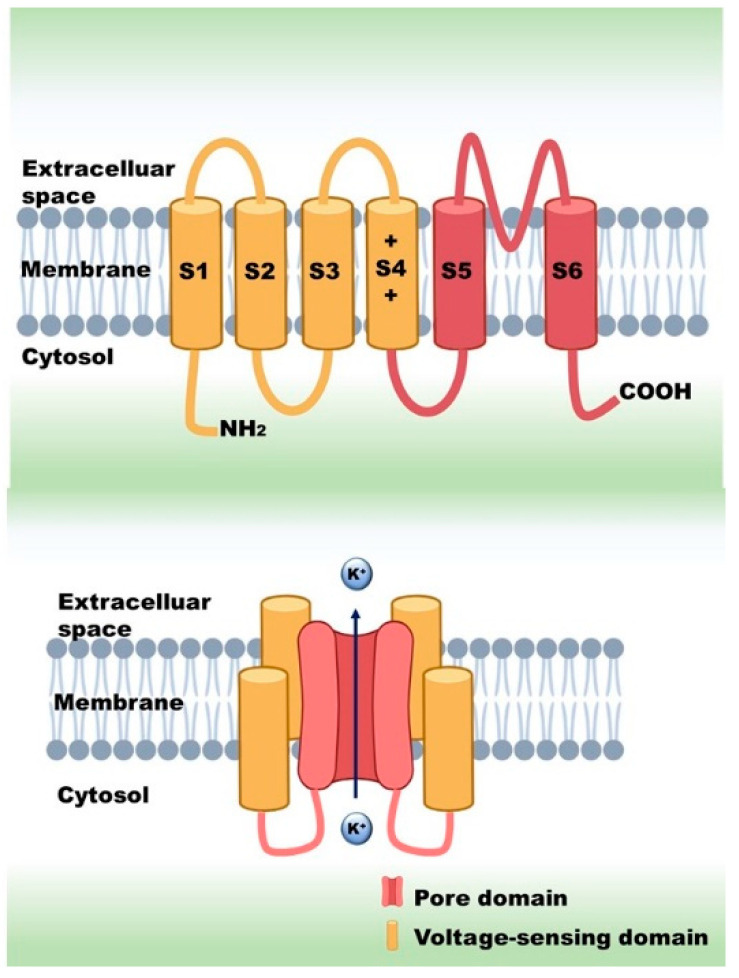
Stucture of voltage-gated potassium channels.

**Figure 2 pharmaceuticals-14-01303-f002:**
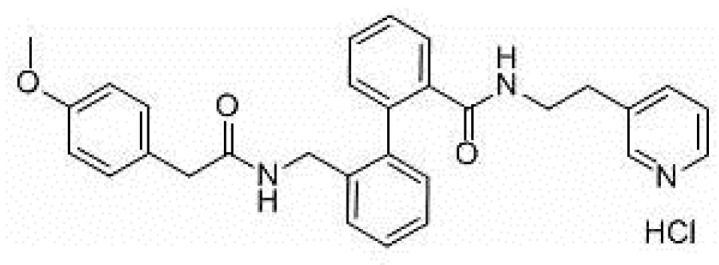
Structure of AVE0118.

**Figure 3 pharmaceuticals-14-01303-f003:**
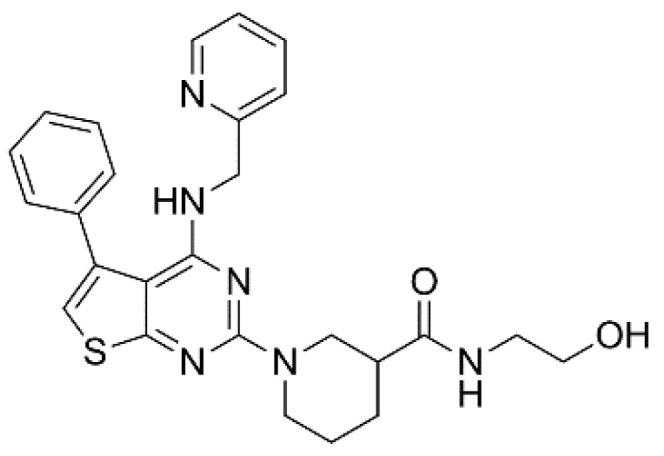
Structure of XEN-D0101.

**Figure 4 pharmaceuticals-14-01303-f004:**
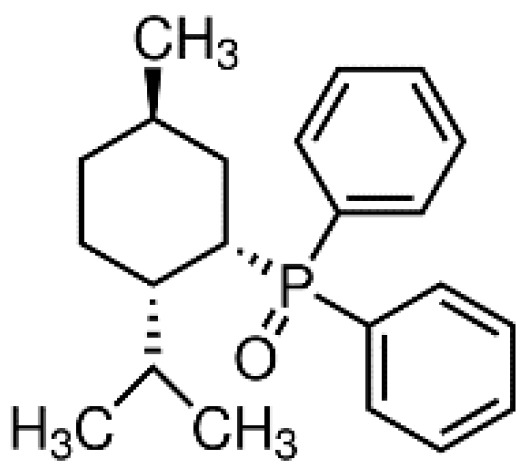
Structure of DPO-1.

**Figure 5 pharmaceuticals-14-01303-f005:**
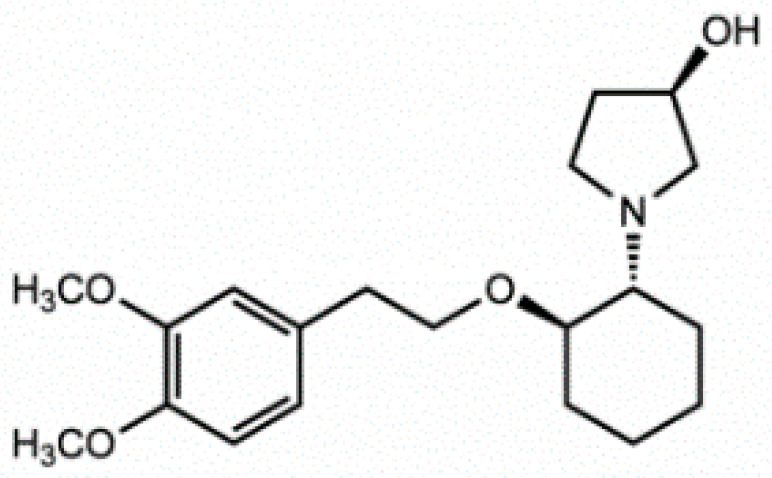
Stucture of Vernakalant.

**Figure 6 pharmaceuticals-14-01303-f006:**
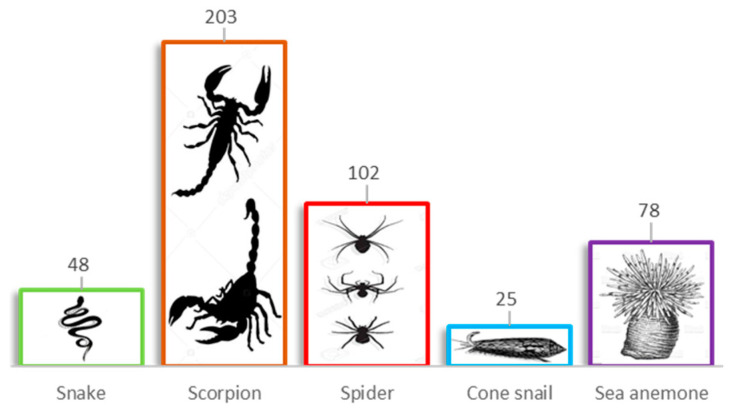
Peptide modulators of the voltage-gated potassium (Kv) channels. Numbers above the boxes indicate the number of toxins isolated to target Kv channels.

**Figure 7 pharmaceuticals-14-01303-f007:**
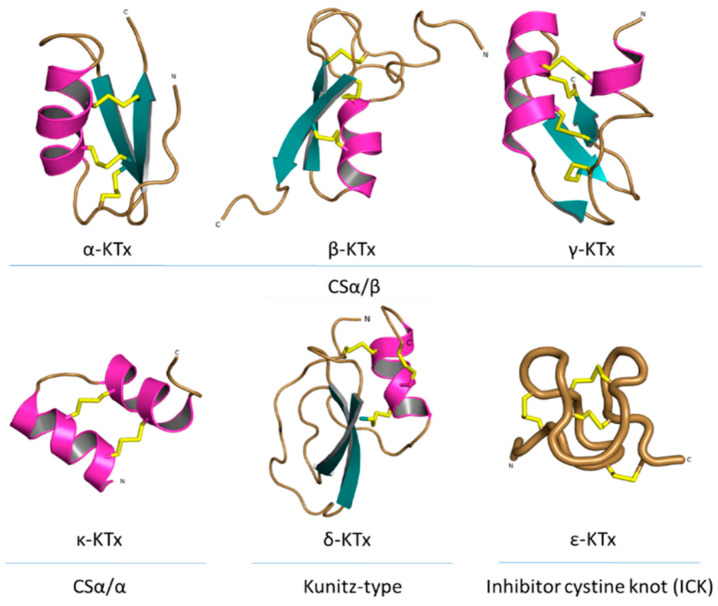
Structural folds present in the KTx families. α-KTx: MgTx (1MTX), β-KTx: HgeScplp1 (5IPO), γ-KTx: Ergtoxin (1PX9), κ-KTx: OmTx2 (1WQC), δ-KTx: LmKTT-1a (2M01) and ε-KTx: Ts11 (2MSF). PDB entries are shown in parentheses.

**Figure 8 pharmaceuticals-14-01303-f008:**
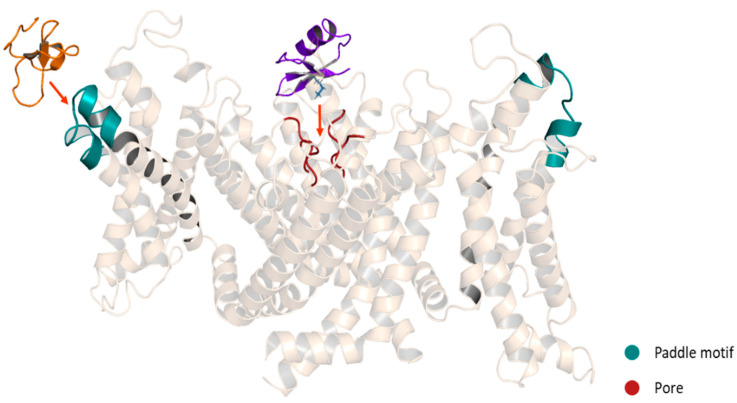
Schematic representation of Kv modulators binding sites. Purple: ChTx, pore blocker. Orange: HaTx1, gating modifier. A chimeric structure of the channels Kv1.2-2.1 is shown (5WIE). Helices in front and behind the structure were removed for a better appreciation.

**Figure 9 pharmaceuticals-14-01303-f009:**
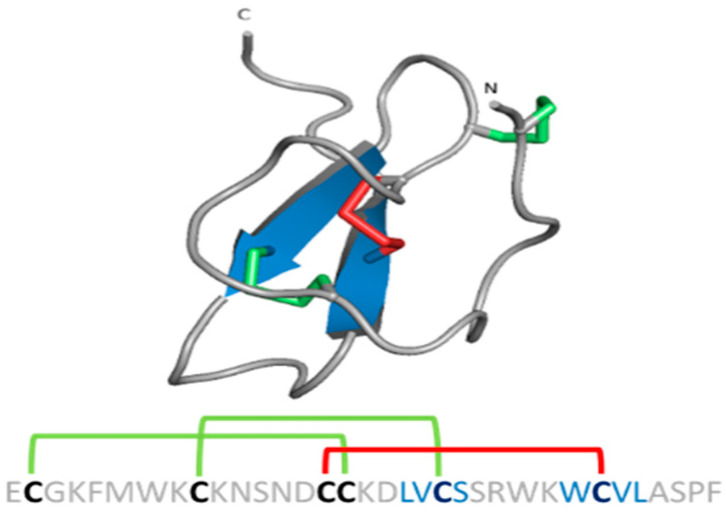
ICK scaffold in peptide gating modifiers. Sequence and structure of the VsTx1 toxin. Disulfide bridge ring is shown in green.

**Figure 10 pharmaceuticals-14-01303-f010:**
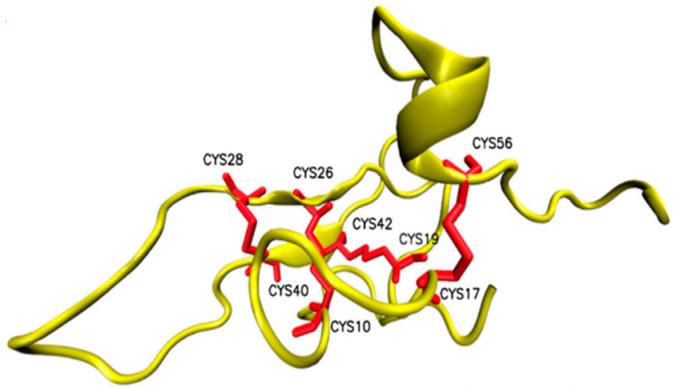
Structure of Osu1 peptide based on homology model.

**Figure 11 pharmaceuticals-14-01303-f011:**
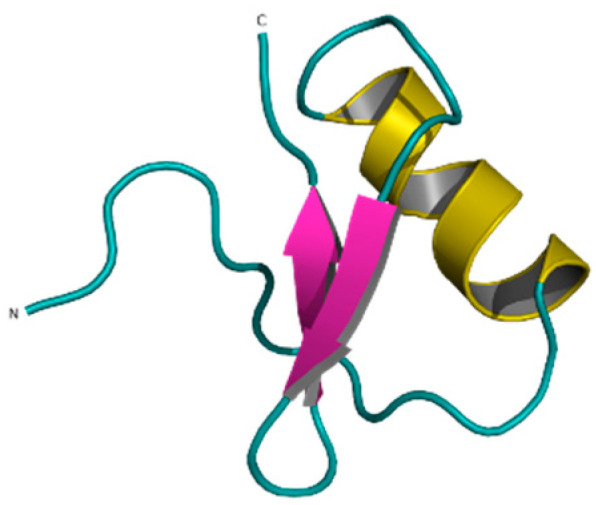
Ts6 toxin (1C56).

## Data Availability

Data sharing not applicable.

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
