# Peer review of "Peptide Inhibitors of Kv1.5: An Option for the Treatment of Atrial Fibrillation"

_pharmaceuticals, 2021, doi:10.3390/ph14121303_

Round 1
Reviewer 1 Report
The authors present a review on Kv channels inhibitors, with a specific focus on peptide inhibitors of Kv1.5 as an option for the treatment of atrial fibrillation. The manuscript is comprehensive and the topic is thoroughly discussed, from ion channels structure to experimental models of atrial fibrillation.
A relevant comment regards section 5. Authors correctly report Ts6 inhibitory activity on Kv1.5. However, Ts6 stimulates the release of NO, IL-6 and TNF-α, and presents a pro-inflammatory activity in experimental models [see: Zoccal KF, et al., Ts6 and Ts2 from Tityus serrulatus venom induce inflammation by mechanisms dependent on lipid mediators and cytokine production, Toxicon, 2013 (already a reference), and Zoccal KF, et al., Tityus serrulatus venom and toxins Ts1, Ts2 and Ts6 induce macrophage activation and production of immune mediators, Toxicon, 2011]. Systemic inflammation, especially IL-6-mediated, is an established risk factor for a number of cardiovascular diseases, including atrial fibrillation [see, for example: Lazzerini PE, et al., Systemic Inflammation Rapidly Induces Reversible Atrial Electrical Remodeling: The Role of Interleukin-6-Mediated Changes in Connexin Expression. J Am Heart Assoc, 2019]. In my opinion, this is a critical and interesting point that should be included in the discussion.
Minor comments only relate to the English – which in general is adequate. In section 3, line 112: consider replacing “sluggish flow” with “turbulent flow”, which is more scientific. Section 3, line 119: “stroke” into “haemorragic stroke”, in order to distinguish from ischemic stroke, which is actually prevented by anticoagulants. Section 3, line 140: “in addition blocking” should be “in addition to blocking”. Section 3, lines 144-145: the sentence “this observation is consistent with a previous study with non-selective IKur blocker 4 aminopyridine” is duplicated, please edit.
A few minor typos should be addressed. In section 5, line 304: “memebrane” should be “membrane”. In section 8, line 461: “pepetides” should be “peptides”.
Author Response
Reviewer #1
Comments and Suggestions for Authors
The authors present a review on Kv channels inhibitors, with a specific focus on peptide inhibitors of Kv1.5 as an option for the treatment of atrial fibrillation. The manuscript is comprehensive and the topic is thoroughly discussed, from ion channels structure to experimental models of atrial fibrillation.
A relevant comment regards section 5.
Authors correctly report Ts6 inhibitory activity on Kv1.5. However, Ts6 stimulates the release of NO, IL-6 and TNF-α, and presents a pro-inflammatory activity in experimental models [see: Zoccal KF, et al., Ts6 and Ts2 from Tityus serrulatus venom induce inflammation by mechanisms dependent on lipid mediators and cytokine production, Toxicon, 2013 (already a reference), and Zoccal KF, et al., Tityus serrulatus venom and toxins Ts1, Ts2 and Ts6 induce macrophage activation and production of immune mediators, Toxicon, 2011]. Systemic inflammation, especially IL-6-mediated, is an established risk factor for a number of cardiovascular diseases, including atrial fibrillation [see, for example: Lazzerini PE, et al., Systemic Inflammation Rapidly Induces Reversible Atrial Electrical Remodeling: The Role of Interleukin-6-Mediated Changes in Connexin Expression. J Am Heart Assoc, 2019]. In my opinion, this is a critical and interesting point that should be included in the discussion.
We would like to thank Reviewer #1 for calling our attention to the pro-inflammatory effect caused by Ts6. We extended our review according to the recommendation.
Minor comments only relate to the English – which in general is adequate.
In section 3, line 112: consider replacing “sluggish flow” with “turbulent flow”, which is more scientific. Section 3, line 119: “stroke” into “haemorragic stroke”, in order to distinguish from ischemic stroke, which is actually prevented by anticoagulants.
Section 3, line 140: “in addition blocking” should be “in addition to blocking”.
Section 3, lines 144-145: the sentence “this observation is consistent with a previous study with non-selective IKur blocker 4 aminopyridine” is duplicated, please edit.
A few minor typos should be addressed.
In section 5, line 304: “memebrane” should be “membrane”.
In section 8, line 461: “pepetides” should be “peptides”.
We have changed the noticed typos and corrected inaccurate words and phrases.

Reviewer 2 Report
The authors describe the human voltage gated potassium channels and the promise of developing selective peptide inhibitors to treat atrial fibrillation. They have done a good job at describing the voltage gated ion channels and their role in several conditions. Furthermore, they have expanded on the some of the inhibitors developed and in vitro and in vivo assay systems that are available to test their potential therapeutic applications.
The authors have done a thorough job of describing voltage gated channels though the review lacks adequate focus on the peptide inhibitors for Kv1.5. They need to further expand on some of the selectivity and computational studies that have been conducted on similar peptides that target ion channels.
Here are some of the minor issues
- The authors keep mentioning selectivity as the major advantage of peptides though most of the peptides mentioned in the review have multiple targets.
- It would add to the importance of peptides as selective agents if the authors could provide an example of the peptide size conferring selectivity vs a small molecule inhibitor.
- For the two peptides mentioned Ts6 and Osu1, further elaboration on strategies to improve the selectivity needs to be added. The authors do mention computational docking experiments though an example is needed to improve the applicability of the strategy suggested.
- Section 6: Improving selectivity and affinity of peptide toxins
- The authors can expand on the idea of using x-ray crystallography or NMR to identify the structural interactions between peptides and ion channels.
- Computational in silico docking can also be added to this section as a method to improve the selectivity
- Additional techniques that allow for a structure activity relationship study of the peptide-ion channel interaction can be included in this section
- Fragment based design could also be explored.
Author Response
Reviewer #2
Comments and Suggestions for Authors
The authors describe the human voltage gated potassium channels and the promise of developing selective peptide inhibitors to treat atrial fibrillation. They have done a good job at describing the voltage gated ion channels and their role in several conditions. Furthermore, they have expanded on the some of the inhibitors developed and in vitro and in vivo assay systems that are available to test their potential therapeutic applications.
The authors have done a thorough job of describing voltage gated channels though the review lacks adequate focus on the peptide inhibitors for Kv1.5. They need to further expand on some of the selectivity and computational studies that have been conducted on similar peptides that target ion channels.
We would like to thank Reviewer #2 for the helpful comments and suggestions. To the best of our knowledge, we have supplemented our review, following the suggestions.
Here are some of the minor issues
- The authors keep mentioning selectivity as the major advantage of peptides though most of the peptides mentioned in the review have multiple targets.
That is correct, peptide inhibitors are not completely selective either. We tried to emphasize that peptides are generally more selective compared to small molecules. Our sentences may have been misleading, therefore we have rewritten Chapter 6 illustrated by concrete examples, to be more accurate.
- It would add to the importance of peptides as selective agents if the authors could provide an example of the peptide size conferring selectivity vs a small molecule inhibitor.
We fully agree with this. Not only one, but several such examples are now mentioned in the extended Chapter 6.
- For the two peptides mentioned Ts6 and Osu1, further elaboration on strategies to improve the selectivity needs to be added. The authors do mention computational docking experiments though an example is needed to improve the applicability of the strategy suggested.
We must also agree with this: mentioning examples can help and improve understanding when describing different methods. In Chapter 7.2. we added examples.
- Section 6: Improving selectivity and affinity of peptide toxins
- The authors can expand on the idea of using x-ray crystallography or NMR to identify the structural interactions between peptides and ion channels.
- Computational in silico docking can also be added to this section as a method to improve the selectivity
- Additional techniques that allow for a structure activity relationship study of the peptide-ion channel interaction can be included in this section
- Fragment based design could also be explored.
We renumbered the chapters and in Chapter 6. and 7 we expanded the ideas, mentioning and citing cryo-EM as additional technique, along with random recombination of proteins tested by phage display.

Round 2
Reviewer 2 Report
The authors have made a good effort at addressing most of the issues that were raised.
Author Response
We would like to thank Reviewer #2 for the helpful comments and suggestions again.